# Detecting the Morphology of Prograding and Retreating Marsh Margins—Example of a Mega-Tidal Bay

**Guillaume C. H. Goodwin** * **and Simon M. Mudd**

School of GeoSciences, University of Edinburgh, Drummond Street, Edinburgh EH8 9XP, UK;
simon.m.mudd@ed.ac.uk
* Correspondence: willgoodwin1201@gmail.com; Tel.: +33-637934599

**Abstract:** Retreat and progradation make the edges of salt marsh platforms their most active features. If we have a single topographic snapshot of a marsh, is it possible to tell if some areas have retreated or prograded recently or if they are likely to do so in the future? We explore these questions by characterising marsh edge topography in mega-tidal Moricambe Bay (UK) in 2009, 2013 and 2017. We first map outlines of marsh platform edges based on lidar data and from these we generate transverse topographic profiles of the marsh edge 10 m long and 20 m apart. By associating profiles with individual retreat or progradation events, we find that they produce distinct profiles when grouped by change event, regardless of event magnitude. Progradation profiles have a shallow scarp and low relief that decreases with event magnitude, facilitating more progradation. Conversely, steep-scarped, high-relief retreat profiles dip landward as retreat reveals older platforms. Furthermore, vertical accretion of the marsh edge is controlled by elevation rather than its lateral motion, suggesting an even distribution of deposition that would allow bay infilling were it not limited by the migration of creeks. While we demonstrate that marsh edges can be quantified with currently available DTMs, oblique observations are crucial to fully describe scarps and better inform their sensitivity to wave and current erosion.

**Keywords:** salt marsh; topographic analysis; progradation; retreat; scarp; erosion; lidar

## 1. Introduction

The alarming landward retreat of well-known salt marsh systems such as the Mississippi's "Bird Foot" [1,2] or the Venice Lagoon [3] has sparked concern for the future of these highly valuable landscapes [4–6]. Salt marshes filter organic and metallic pollutants [7,8] and provide important nursing grounds for wildlife, including commercially exploited species such as Brown Shrimp [9]. Furthermore, their high productivity makes salt marshes important sites of blue carbon sequestration [10] and their vegetation and topography reduce storm surges and damp waves [11–15]. The loss of salt marshes to the sea is predicted to cause significant losses to the ecosystem services they provide [16] and release stored carbon into the ocean [17,18], diminishing its capacity to siphon atmospheric carbon.

Although the extent of their vulnerability is regularly debated [19–21], studies repeatedly show that some salt marsh environments are at risk of drowning due to sea level rise [22,23] despite the bio-geomorphic feedbacks [24,25] that led to the emergence of marsh platforms from bare mudflats in the first instance [26]. Frequently, this drowning has been attributed to insufficient sediment supply [27,28].

Vertical challenges to salt marsh survival are matched by lateral retreat, notably driven by waves and tidal currents. Multiple studies have focused on the impact of external forcing on the landward

constriction of salt marsh habitat [29–31], as well as the mutual interaction between wave impact, retreat processes and the morphology of retreating marsh margins [32–34]. While marsh retreat is demonstrably linked to nearby channel deepening in a macro-tidal setting [35,36], the action of tidal currents on marsh margins remains poorly understood relative to wave action.

Likewise, remote observation of salt marsh margins are scarce in the literature, in contrast with the wealth of documentation on the use of light detection and ranging (lidar) and hyperspectral data to characterise marsh platform elevation and vegetation [37–40]. This knowledge gap hampers our understanding of present coastal mobility in general but also our predictions of the future retreat or advance (which we refer to as progradation) of salt marshes. The mobility of marsh edges is often studied through the determination of wave- or current-generated stresses rather than direct observation of marsh edges. This lack of observation data prevents us from contextualising results on the influence of scarp topography on wave action [32].

The paucity of data on marsh edge topography may be due to technical difficulties: in many micro-tidal systems and some meso-tidal systems the foot of the marsh scarp is rarely exposed [41] and few sites have as good topo-bathymetric data as the repeatedly studied Venice Lagoon in Italy [42] and Plum Island in Massachussets, USA [43], both of which are the object of long-term monitoring campaigns. Moreover, the spatial resolution of airborne lidar images is usually in the range of 1–5 m, which reduces the perceived slope of scarps, despite being the most fine-grained remote sensing method used to cover large marsh systems [44]. More importantly, scarps cannot be observed by nadir-facing airborne lidar surveys due to their sub-vertical face. Finally, many salt marsh are dominated by *Spartina alterniflora* or *Spartina anglica*, plants that lead to errors of 15–55 cm on lidar elevations, with errors of up to 1.70 m along creek banks [45]. For low-lying micro-tidal marshes (and to a lesser extent, meso-tidal marshes), such errors are of the order of scarp heights. These factors combined complicate the study of marsh margin morphology.

Conversely, macro- to mega-tidal mudflats are more frequently exposed, increasing the opportunities for purely topographic surveys. In these conditions marsh platforms are often higher in the tidal frame than their microtidal cousins [46], with retreating margins often taking the shape of scarps more than 1*m* in height fronted by degrading fallen blocks, locally known as saltings (Figure 1c). These scarps contrast sharply with prograding margins, which exhibit shallow or non-existent scarps fronted by pioneer species like *Salicornia* sp. or *Sarcocornia* sp. (Figure 1b). As illustrated by these images of Skinburness Marsh in Moricambe Bay (Cumbria, United Kingdom), grazed marshes dominated by *Puccinellia maritima* have low vegetation near their margin, thus reducing the typical elevation bias caused by vegetation cover [47–49].

Under such conditions, we hypothesise that salt marsh margins are sufficiently well defined to discern their morphology with lidar data. In this contribution, we use modern feature detection methods to extract salt marsh outlines from three lidar surveys covering the sheltered mega-tidal Moricambe Bay. From these outlines, we produce regularly spaced transverse profiles of the marsh margin topography. The profiles are attached to unique change events corresponding to localised and contiguous retreat or progradation between two observation times. Using these data, we detach margin profiles from their spatial context to examine the morphological difference between retreating and prograding margins. We then focus on the properties of marsh margin relief to examine the distinctive properties of prograding and retreating margins with respect to the volume of displaced sediment, as well as the response of marsh margin elevation to retreat or progradation. The variety of retreating and prograding marsh margins in Moricambe Bay allows us to examine the morphology of a wide range of active margins for the years 2009, 2013 and 2017.

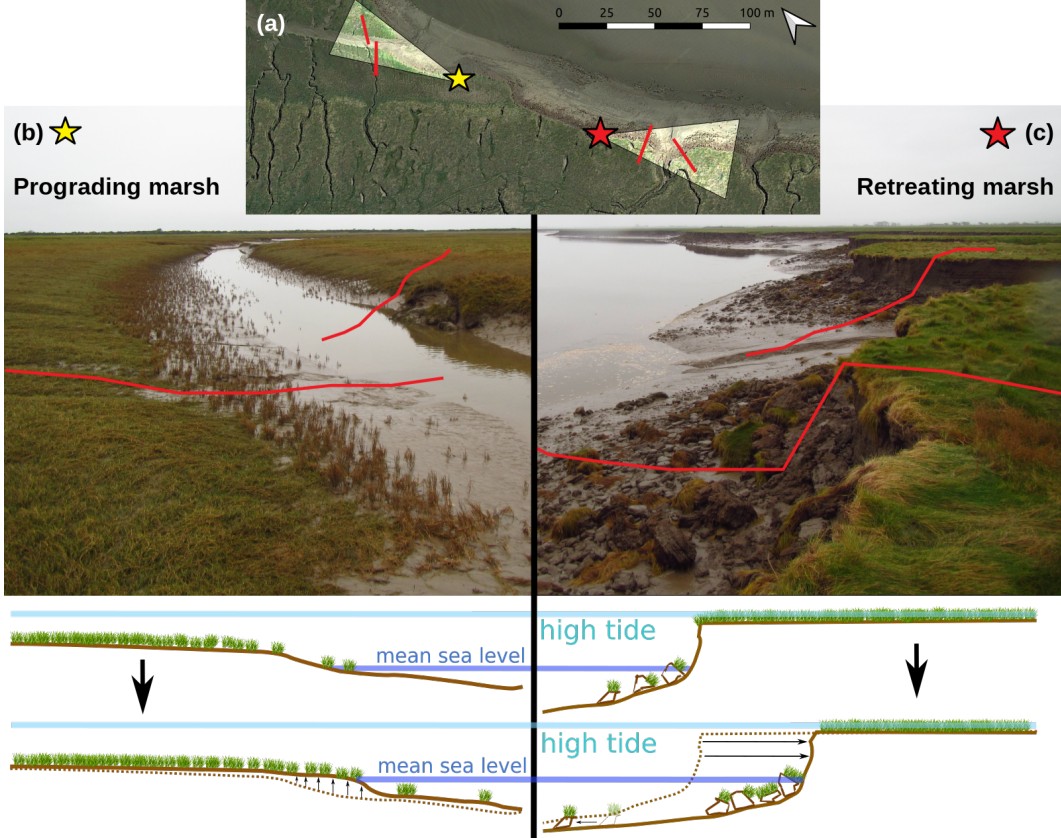

**Figure 1.** Aerial and ground views of salt marsh margin profiles. (**a**) aerial view of photography point of views and location of profiles; (**b**) photography of prograding margins (G. Goodwin, November 2016) and schematic profiles. The diagramme below provides a schematic view of the process of progradation; (**c**) photography of retreating profiles (G. Goodwin, November 2016) and schematic profiles. The diagramme below provides a schematic view of the process of retreat. Profiles on the photographs in panels (**b**,**c**) are deliberately drawn with low resolution to illustrate the perspective from 1*m* lidar data.

## 2. Site Description

The Solway Firth is a mega-tidal estuary separating Dumfries and Galloway in Scotland from Cumbria in England (Figure 2b). The Northern Cumbrian coastline is renowned for its active salt marshes, which show evident signs of both retreat and progradation (Figure 1). The bay of Moricambe, its North-West facing entrance enclosed between the Grune Cast sand spit and Cardurnock Flatts, is no exception and provides a sheltered environment where wide marshes have developed (Figure 2a). There, the meandering of the tidal rivers Wampool (North) and Waver (South) appear to be the main constraint on the development of salt marshes, generating autocyclic retreat and progradation [50], of which the terracing of Skinburness Marsh (see Figure 5a) is a remnant. Likewise, the southern part of Newton Marsh shows signs of progradation enabled by the further distance of channels.

Such diversity in the active marsh margins is central to our study. The main activity on the salt marshes is cattle grazing, with both dairy cows and sheep regularly being kept in pastures on the marsh platforms. Hence, the dominant vegetation in Moricambe Bay is grazed *Puccinellia maritima* which seldom exceeds 1–5 cm in height. This makes it an ideal site upon which to study marsh evolution using high resolution topographic data, as the low vegetation minimizes errors in topographic data. High resolution lidar topography covering the whole of Moricambe Bay is freely available through the UK Department for Environment and Rural Affairs (DEFRA), allowing for the implementation of feature-based marsh platform detection.

## 3. Materials and Methods

### 3.1. Collection and Pre-Processing of Topographic Data

We download point cloud topographic data from airborne lidar surveys of Moricambe Bay within the area of interest (red polygon in Figure 2a) from the DEFRA data repository for 2009, 2013 and 2017 (https://environment.data.gov.uk/DefraDataDownload/?Mode=survey). DEFRA provides the last return for every point (the density of which does not exceed 6 pts·m$^{-2}$). This does not necessarily imply that the last return is the ground or bare earth, as dense vegetation on the marsh platform may prevent the laser from hitting the ground [49,51,52]. However, thanks to pastoral activities in Moricambe Bay, vegetation rarely exceeds 5 cm and does not cause significant errors in measured elevations such as those reported reported by Hladik and Alber [51] on marshes with tall vegetation. We convert the point clouds to rasters with a grid resolution of 1 m, generating Digital Terrain Models (DTMs) for each year. At the ground-truthing points within the Ordnance Survey tile NY15 (https://environment.data. gov.109uk/DefraDataDownload/?mapService=EA/LIDARGroundTruthSurveys&Mode=spatial) , we find that the mode of the 2017 DTM is higher than the mode of the 2013 DTM by 7 cm and than the mode of the 2009 DTM by 5 cm (Figure A1).

For the purposes of this contribution, we are more interested in short-term sediment deposition or removal than long-term land movements caused by post-glacial uplift. We correct for long-term land movements by comparing stable infrastructure (e.g., roads) between DTMs. For these corrections, we use the 2017 DTM as reference. After correction, the Root Mean Square Error (RMSE) between GPS-acquired points and the lidar DTM are the following: for the 2009 DTM, the RMSE is 6.8 cm (Figure A2a); for the 2013 DTM, the RMSE is 6.5 cm (Figure A2b); for the 2017 DTM, the RMSE is 3.1 cm (Figure A2c). Each DTM is then clipped to the area of interest illustrated in Figure 2a. Because of the low vegetation shown in Figure 1, we do not apply an additional elevation correction to account for vegetation on the salt marsh platforms.

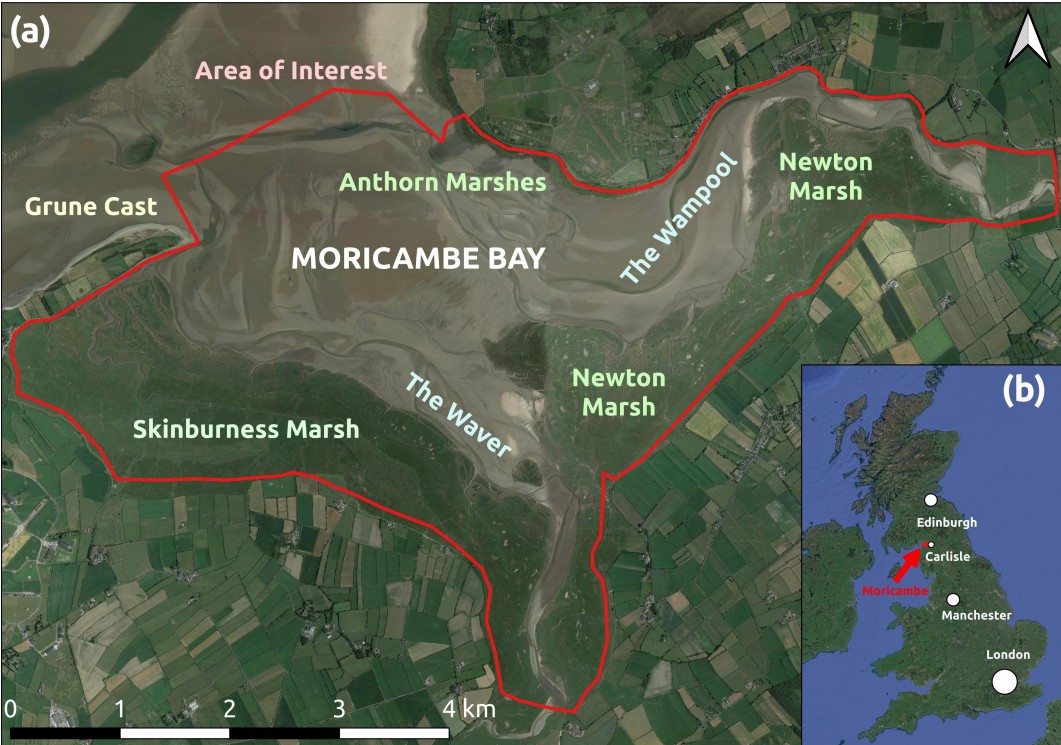

**Figure 2.** Satellite view of Moricambe Bay (**a**) and location within the United Kingdom (**b**). Image credit Google Earth (30 June 2018).

### 3.2. Determination of Marsh Outlines and Profiles

For each of the DTMs, we isolate marsh platforms using the Topographic Identification of Platforms (TIP) method [53]. The TIP method uses a high-resolution DTM in raster format (e.g., from lidar data) to classify pixels as "marsh platform" or "tidal flat" within an area of interest. The TIP method proceeds in two major steps: (i) the determination of marsh outlines and (ii) the filling of marsh platforms.

For step (i), the product of dimensionless relief and slope is calculated as shown in Equation (1):

$$P^* = \frac{z - z_{min}}{z_{max} - z_{min}} * \frac{s - s_{min}}{s_{max} - s_{min}}, \tag{1}$$

where $z$ is the elevation of the pixel, $z_{min}$ is the minimum elevation in the DTM and $z_{max}$ is the maximum elevation in the DTM. The same notation applies to the pixel slope $s$, determined from the DTM after [54]. The distribution of $P^*$ in a DEM is exponentially decreasing: hence, pixels for which the slope of the distribution of $P^*$ is lower than $Sp_{thresh}$ are retained as potential marsh scarps. Local maxima of $P^*$ are then used to initiate scarps, which are then routed along "crests" of high $P^*$. $ZK_{thresh}$ then determines a high-pass filter to determine definitive scarps. This step is sensitive to the presence of small marsh scarps. For step (ii), platforms are generated by progressively "filling" the pixels above the scarps over multiple iterations. Pixels in the lower part of the elevation distribution of the newly generated platforms are then eliminated, using $rz_{thresh}$ to determine the percentage of the distribution to eliminate after the lowest point of the elevation distribution. The result of these two steps is a classified raster, with values of 0 for tidal flats and 1 for marsh platforms.

Moricambe Bay is larger than most sites for which the TIP method was tested. Furthermore, while the TIP method was shown to be effective for marsh platforms exhibiting a well-defined scarp, this is not the case everywhere in Moricambe Bay. Hence, we separate the study site into 21 sectors and implement the TIP method on each sector with different parameters. The sectors were determined to minimise the overlap of mature and young platforms within any given sector, so as to avoid the TIP method mistaking the younger, lower platforms for tidal flats. Figure A3 shows the layout of the sectors and Tables A1, A2 and A3 record the parameters used in each sector. The TIP method tends to exclude pools and disconnected channels from the marsh platform, thus creating complex and discontinuous marsh platforms which do not correspond to the most seaward marsh margin. To keep only the most seaward outlines, we invert the TIP method's original results (see Figure A4a) to identify tidal flats, of which we select only the largest. In Figure A4a, this is the northernmost tidal flat. Any pixel within the area of interest not classified as a tidal flat is then considered a marsh platform, yielding Figure A4b. A close-up of marsh platforms for each year are shown in green in Figure 3a.

Along the seaward outline of each marsh platform, we generate transverse profiles of 10 m in length, spaced regularly by 20 m, as shown in Figure 3a. Each 10 m long profile contains 11 vertices (one each meter, including the starting and ending points). We extract the topography of each individual profile for all 3 years, as shown in Figure 3b–d. Each year will have its own set of marsh profiles. This is because the orientation of the marsh edge changes when the marsh outline prograde or retreats: hence, a profile that is orthogonal to the marsh outline in 2009 may not be in 2013 or 2017, thus rendering a direct comparison of profile geometry impossible. An approach using sets of profile for each year is therefore preferable to one using a single set of profiles for all three years. Indeed, the latter approach, using longer profiles, would be suited to analyse the geometry of entire marsh platforms but not of features with small footprints like scarps. But in addition we record the elevations at every profile vertex for all three years. That means that any set of 11 nodes within an individual year's profile will be associated with 3 topographic profiles.

Each vertex $p_i$ of a profile $p$ is defined by the coordinates $(p_{i,x}, p_{i,z})$, respectively the seaward distance and elevation of $p_i$. The marsh edge $p_{ma}$ is defined as the first 4 vertices of $p$ (green background in Figure 3b–d), while the mudflat edge $p_{mu}$ is defined as the last 4 vertices of $p$ (brown background). We introduce this subdivision of the profiles to avoid the influence of fallen blocks when determining the relief $R$, defined in Section 4.3. In the example shown in Figure 3, profiles in 2009 and 2013 show

little signs of a scarp (b,c), hinting at a prograding evolution which is stopped in 2017, as we observe a visible retreat scarp about 1 m further inland than the scarps in 2013 and 2009 (d).

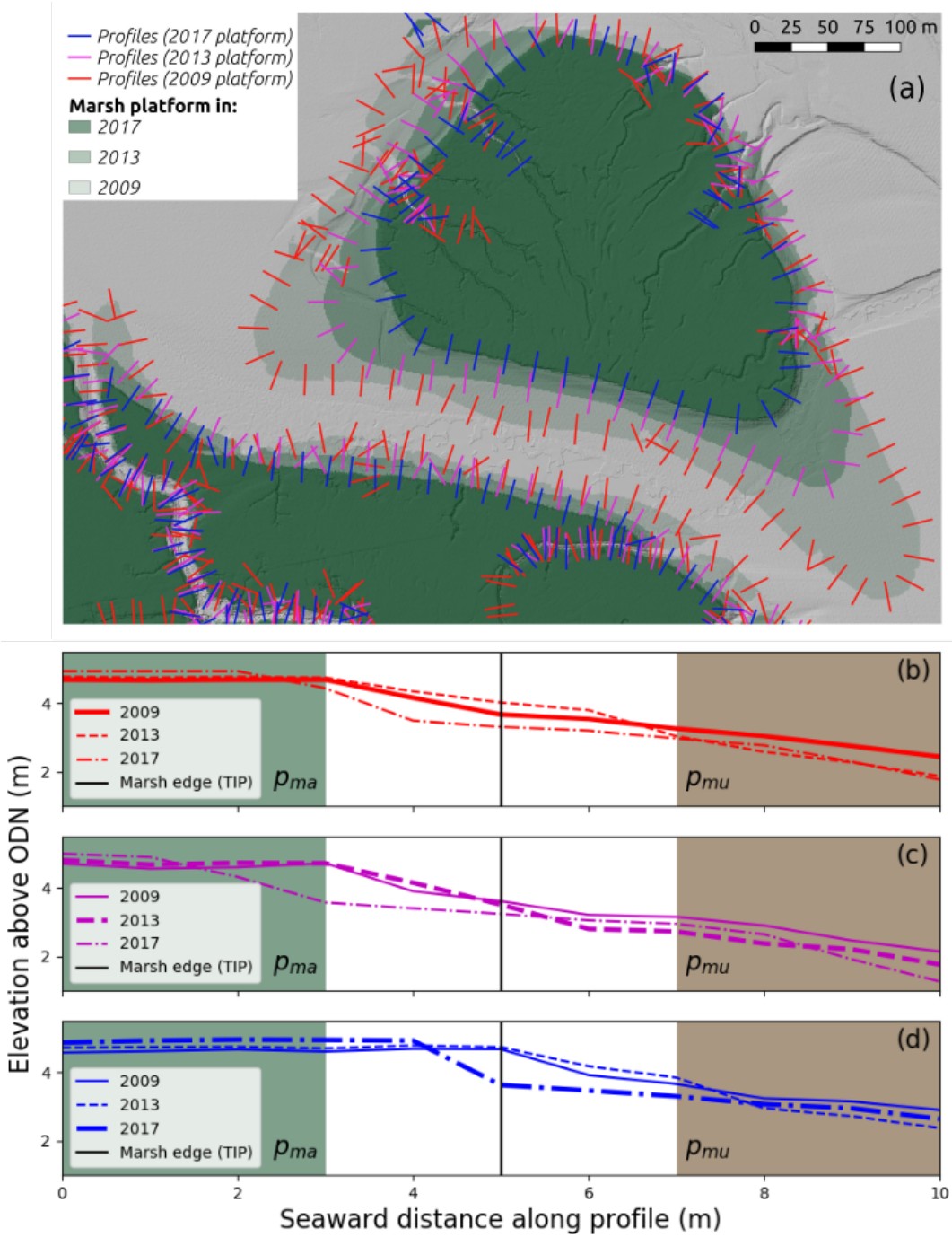

**Figure 3.** Evolution of scarp profiles over the years: (**a**) map of marsh platforms near the mouth of the Waver and location of scarp profiles; example of a scarp profile associated with a various marsh outlines, with elevations for all three years; (**b**) 2009 outline; (**c**) 2013 outline; (**d**) 2017 outline. Bold lines indicate the current profile. In (**b–d**) green portions represent the marsh-side of the profile $p_{ma}$ and brown portions represent the mudflat-side of the profile $p_{mu}$.

### 3.3. Determination of Change Events

By comparing the marsh platforms generated in Section 3.2, we determine the trajectory of each pixel between 2009 and 2017, defined as the record of its classification as marsh platform or mudflat.

Each of the 8 possible trajectories for a pixel is shown in Figure 4b. For instance, a pixel classified as a marsh platform in 2009, as a mudflat in 2013 and as a marsh platform in 2017 would follow trajectory 8. The trajectory of each pixel as seen in Figure 4a is colour-coded according to Figure 4b. All pixels except those following trajectories 1 and 2 undergo at least one change of classification between 2009 and 2017.

As illustrated in Figure 4a, groups of contiguous pixels tend to follow the same trajectory. Even if pixels do not share a full trajectory, many share partial trajectories. For instance, pixels following trajectories 4 and 8 are both converted to mudflats between 2009 and 2013. In this contribution, we refer to groups of contiguous pixels this conversion as change events ($CE$), indicated as red and blue circles in Figure 4b. In this instance, a change event involving the conversion of marsh platforms to mudflats and occurring between 2009 and 2013 may include pixels following trajectories 4 and 8. Likewise, a change event involving the conversion of mudflats to marsh platforms and occurring between 2013 and 2017 will include pixels following trajectories 8 and 5. Thus, a pixel may be involved in up to two change events and each change event is a unique group of contiguous pixels that can be given a unique identification.

We identify all change events larger than 3 contiguous pixels (3 m$^2$), with contiguity being defined within neighbourhoods composed of the eight adjacent pixels (i.e., both cardinal directions and diagonal pixels). Retreat events ($RE$), during which the marsh margin recedes landward, are lined with the most recent profiles on the landward side and the least recent on the seaward side and vice versa for progradation events ($PE$). Thus, each change event accepts as boundaries the marsh outlines that border it and is associated with two sets of profiles: one preceding the change and another resulting from the change (Figure 4c). This association between change events and sets of profiles will constitute the basis of our morphological analysis.

Individual change events in each of the 2009–2013 and 2013–2017 periods can be quantified by their total volume $V_{CE}$, surface area $A_{CE}$ and average sediment accumulation $h_{CE}$. Throughout this contribution, we show volumes of change events as positive values to accommodate logarithmic scaling in our figures. However, since retreat events are associated with loss of sediment, change event volume in the figures is such that $V_{CE} = V_{PE}$ for progradation events and $V_{CE} = -V_{RE}$ for retreat events.

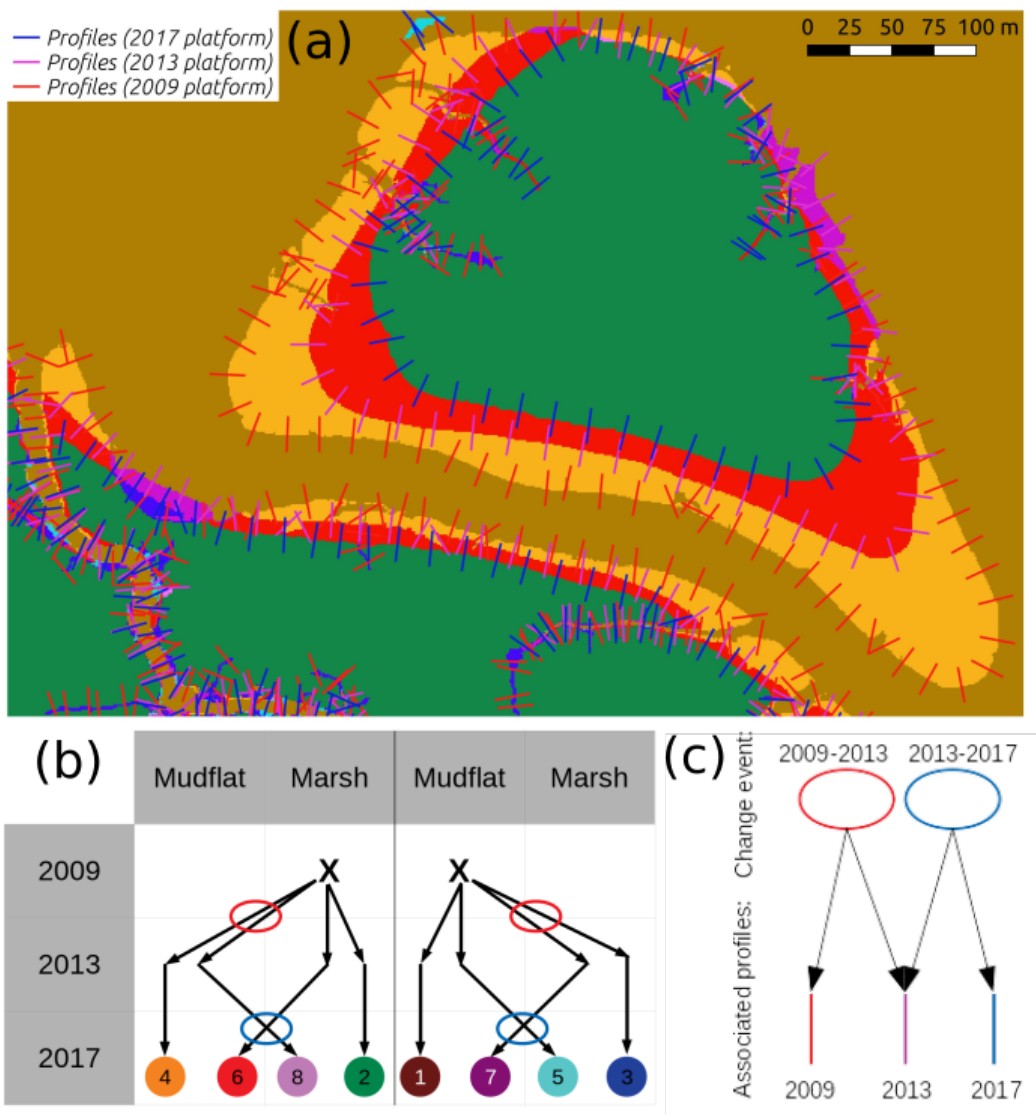

**Figure 4.** Diagram showing possible change events: (**a**) map of classified pixel trajectories near the mouth of the Waver and location of scarp profiles; (**b**) colour and number codes for each of the 8 possible pixel trajectories. Ellipses represent retreat or progradation events for each trajectory; (**c**) Diagram showing how change events are associated with profiles. Example: in (**a**), 6 contiguous areas are coded 4, thus generating 6 individual change events; in (**b**), pixel trajectories coded 4 mark retreat between 2009 and 2013 followed by stability; in (**c**), the profiles at the boundaries of the areas coded 4 are 2009 on the seaward side and 2013 and 2017 on the landward side, since there was no evolution between 2013 and 2017.

### 3.4. Profile Comparison and Metrics

In order to understand to what extent change events are correlated with the geometry of marsh margins, we investigate the differences between prograding and retreating profiles. Margin profiles are grouped in sets, each set being associated with a unique change event. To compare the morphology of margin profiles, we define the mean absolute elevation difference $\Delta_{P,N}$ of a set $P$ of $N$ profiles each of length $L$ in Equation (2):

$$\Delta_{P,N} = \frac{2}{N(N-1)} * \sum_{\substack{k=1 \\ j<k}}^{N} \sum_{j=1}^{N} \frac{\sum_{i=1}^{L} \sqrt{((pj_{i,z} - pj_{0,z}) - (pk_{i,z} - pk_{0,z}))^2}}{L}, \tag{2}$$

where $(pj_{i,z} - pj_{0,z})$ is the elevation of the vertex $pj_i$ of the profile $pj$ relative to the elevation of the first vertex $pj_{0,z}$. The first sum defines the average geometric difference between two profiles by comparing them relatively to their respective most landward elevation. The term $\frac{2}{N(N-1)} * \sum_{k=1}^{N} \sum_{\substack{j=1 \\ j<k}}^{N} X$ is the average of the first sum over all possible combinations of non-identical profiles within $P$. For example, a set $P$ for which $\Delta_{P,N} = 0$ would contain profiles of identical geometry, regardless of their location.

For small events, the close proximity of profiles may play a role in their similarity. Numerous small events may then skew the distribution of $\Delta_{P,N}$ toward lower values. To test this hypothesis, we define the mean inter-profile distance $D_{P,N}$ of a set $P$ of $N$ profiles in Equation (3):

$$D_{P,N} = \frac{2}{N(N-1)} * \sum_{k=1}^{N} \sum_{\substack{j=1 \\ j<k}}^{N} d_{jk}, \tag{3}$$

where $d_{jk}$ is the distance between the centroids of profiles $pj$ and $pk$. $D_{P,N}$ therefore expresses the average distance between pairs of profile centroids within a change event. We also define metrics to characterise profile geometry in simple terms: first, the relief $R$ is defined for a set $P$ of $N$ profiles according to Equation (4), where $p\tilde{m}_{a,z}$ is the median elevation of the marsh portion of all profiles in the set and $p\tilde{m}_{u,z}$ is the median elevation of the mudflat portion of all profiles in the set.

$$R = \tilde{p}_{ma,z} - \tilde{p}_{mu,z}. \tag{4}$$

We also define the maximum Slope $Smax$ of the marsh scarp according to Equation (5), where $i \in [3:6]$ and $l = 1m$. This definition ensures that $Smax$ is the closest possible approximation of the real scarp slope.

$$Smax = max\left(\frac{p_{i,z} - p_{i+1,z}}{l}\right). \tag{5}$$

Finally, we define the slope $S$ of the marsh platform and the mudflat according to Equation (6), where $i = 0, j = 3, l = 3m$ (marsh slope $S_{ma}$) and $i = 7, j = 10, l = 3m$ (mudflat slope $S_{mu}$).

$$S = \frac{\widetilde{p_{i,z} - p_{j,z}}}{l}. \tag{6}$$

## 4. Results and Discussion

### 4.1. Location and Properties of Change Events

The elevations of Moricambe Bay in 2009 can be seen in Figure 5a, where we show marsh platforms in colour and mudflats in greyscale. Most marsh platforms have an elevation range of 4–7 m. However, western Skinburness Marsh shows visible terracing, indicating a progressive development of the marsh in the shadow of Grune Cast with multiple growth interruptions [55]. In this case, the interruptions were caused by the meandering of the River Waver [50]. Both Skinburness and Newton Marshes show a distinctive increase in elevation with distance upstream of the tidal rivers, indicating a constriction of tidal flows [56]. By 2013, large progradation events have considerably increased the surface area of Newton Marsh (Figure 5b), depositing 1 m or more of sediment in some areas. Conversely, Skinburness Marsh has receded under the pressure of the meandering Waver, as have the northernmost portions of Newton Marsh under the influence of the Wampool. We note that most outlines that experienced retreat from their 2009 position are bordered by tidal channels in 2013. Marginal progradation is observed on the Anthorn Marshes. By 2017, Skinburness Marsh has retreated even further under the continued migration of the Waver, while the newly formed marshes of Newton Marsh, well advanced within the bay, are more exposed and show mixed behaviour (Figure 5c). This may be attributed to the anabranching of the Wampool along the northern Newton Marsh.

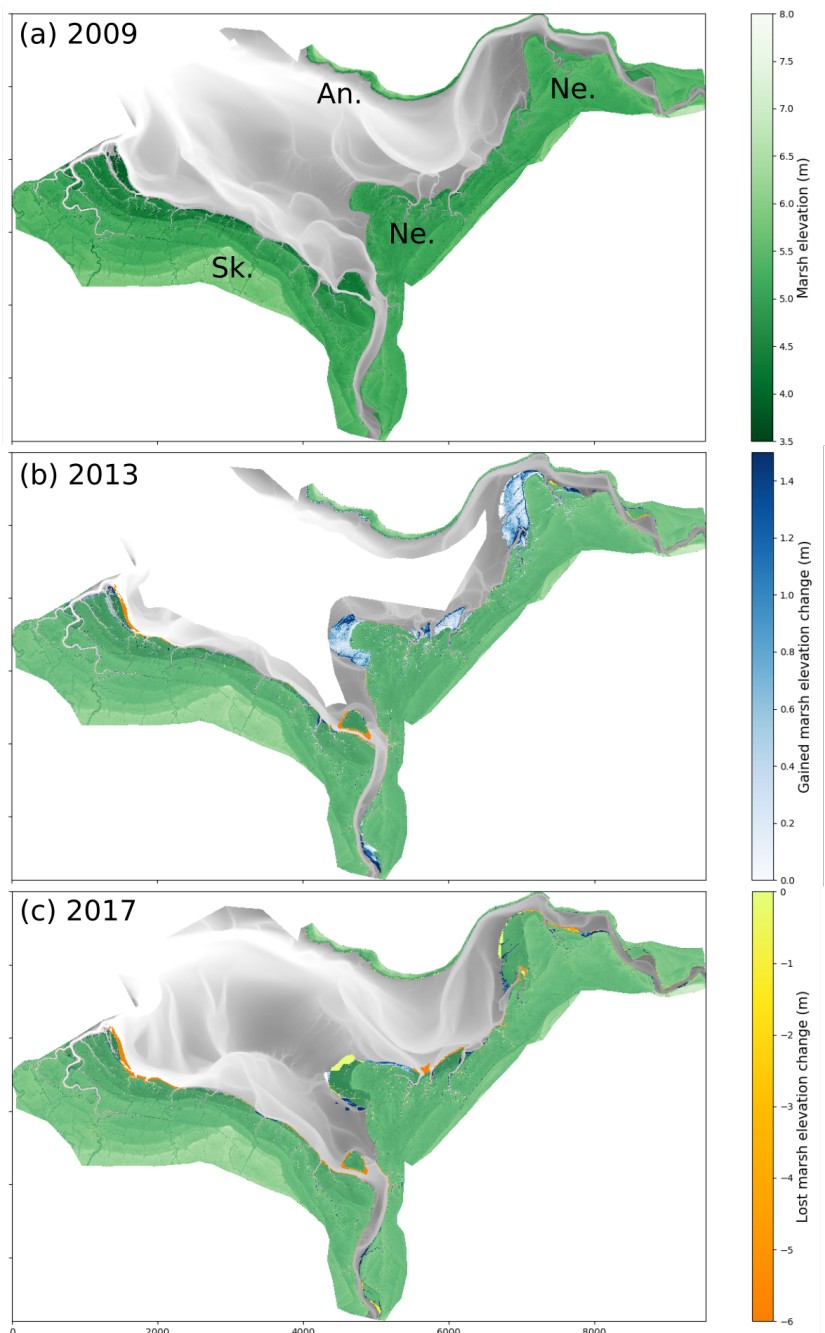

**Figure 5.** Elevation and evolution of Moricambe Bay: (**a**) elevation of marsh platforms (greens) and mudflats (greys) in 2009; (**b**) elevation of marsh platforms and mudflats in 2013 and gained (blues) and lost (oranges) marsh platforms in 2013; (**c**) elevation of marsh platforms and mudflats in 2013 and gained (blues) and lost (oranges) marsh platforms in 2017.

In Figure 6a,c, we observe that the surface-area-to-volume ratio $\frac{1}{h_{CE}} = \frac{A_{CE}}{V_{CE}}$ for progradation is larger than for retreat: indeed, between 2009 and 2013, the two largest retreat and progradation events have similar volumes ($\approx 2 \times 10^4$ m$^3$ and $\approx 4 \times 10^4$ m$^3$). This is in stark contrast with the change in surface area, which for the progradation events is approximately ten times larger. The same trend is observed between 2013 and 2017, although we notice a decrease in the volume and surface area of the largest progradation events. Hence, progradation events deposit less sediment than is eroded during retreat events of the same surface area.

While Figure 6a,c seems to show a linear relationship between Figure $V_{CE}$ and $A_{CE}$, panels (b,d) show that $h_{CE}$ appears to be nonlinearly related to the volume of change. The rate of $h_{CE}$ increases with increasing $V_{CE}$ for retreat events, hinting that larger retreat events may be caused by the migration of deeper creeks or the retreat of higher marsh platforms. During the largest retreat events, which correspond to the migration of the Waver into Skinburness Marsh, approximately 4 m of elevation is removed on average, showing the conversion of a reasonably high marsh platform into a deep tidal creek and not a tidal mudflat.

In prograding marshes, $h_{CE}$ increases very slowly with increasing volume of change, only once exceeding 1 m in depth and averaging under 0.3 m. These rates of accretion remain high but are neither impossible [46] or unheard of for mega-tidal environments with high sediment supply [57].

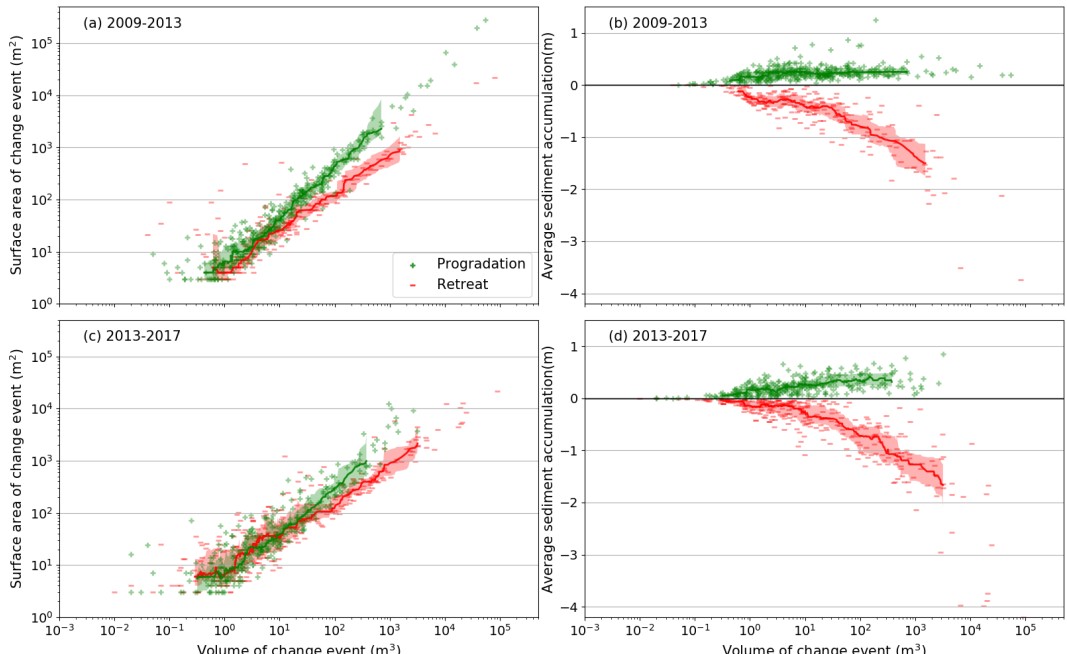

**Figure 6.** Properties of change events, expressed as a function of change event volume (the volume of loss events is negative). Surface area gained (green) or lost (red) in the 2009–2013 period (**a**) and 2013–2017 period (**c**); Average sediment depth deposited (green) or eroded (red) in the 2009–2013 period (**b**) and 2013–2017 period (**d**). Thick lines are a running median over 30 elements, surrounded by the 1st and 3rd quartiles (filled).

*4.2. Geometric Separation between Retreat and Progradation Profiles*

Figure 7 shows values of $\Delta_{P,N}$ for various cases in groups of six box and violin plots. Each violin plot, within each group, represents the distribution of $\Delta_{P,N}$ for the profiles described in the group. Likewise, boxplots show the first and last ten percentiles (black horizontal line), first and third quartiles (boundaries of the box) and median (orange line) within the distribution illustrated by the violin plots. The first and third groups focus on profiles in 2009 and 2013 about to be affected by change events, while the second and fourth group focus on profiles resulting from change events in 2013 and 2017. Within each group, solid line violin plots and their associated boxplots show the distribution of $\Delta_{P,2}$ for all pairs of retreating profiles (red), prograding profiles (green) or mixed pairs (grey). Dashed lines show the distribution of $\Delta_{P,N}$ for all sets of profiles tied to a retreat or progradation event (respectively red and green). The final (grey) element of each group represents the distribution of $\Delta_{P,N}$ for all combinations of one retreat event to one progradation event. The first three plots within each group are comparisons amongst pairs of all profiles, whereas the second set of three plots within each group are profiles compared amongst other profiles in their change event. We do this to see if there are universal

differences in the profiles regardless of the change event (the first three plots) and if profiles within a change event or paired change events are similar (the second three plots within each group).

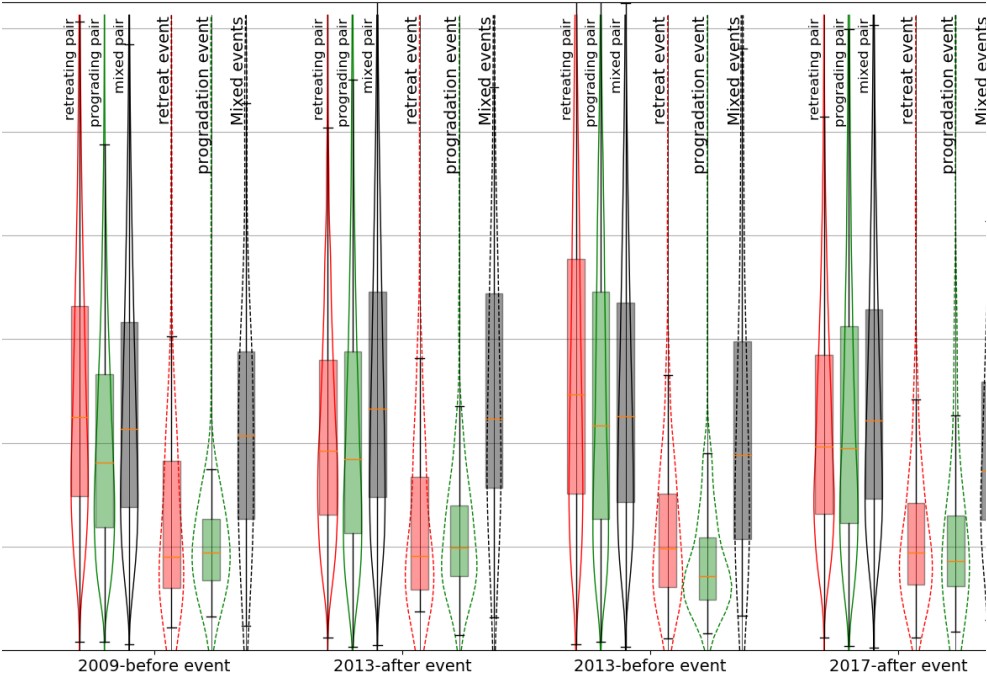

**Figure 7.** Boxplots and full distribution of $\Delta_{P,N}$ for various configurations. Distributions with continuous lines are $\Delta_{P,2}$ for pairs containing two retreating (red) or prograding (green) profiles or mixed pairs (grey). Distributions with dashed lines are $\Delta_{P,N}$ for all retreat events (red) and progradation events (green) or paired retreat and progradation events (grey).

We observe that the distributions of $\Delta_{P,2}$ between retreating, prograding and mixed pairs are not obviously separable, indicating that the morphology of individual profiles alone is not enough to determine whether a profile has undergone or will undergo retreat or progradation. This result appears to contradict accepted understanding that retreating marsh margins exhibit a visible scarp while prograding margins often do not [58,59] (see Figure 1). This is in fact a spurious byproduct of the gridding process from airborne lidar: DTMs derived in this way offer a nadir-facing perspective that cannot detect the near-vertical surfaces that are erosion scarps. Furthermore, aerial lidar data in our case study are gridded with a 1 m$^2$ cell size, which is larger than the typical footprint of a marsh scarp. Hence, the apparent slope of the scarp on a DTM is limited by the cell size of the DTM and is in effect the difference in elevation between two contiguous pixels containing the scarp. This discrepancy is the reason why the TIP method used to determine the marsh outline constructs lines of local slope maxima to locate marsh scarps and variably places the limit of the marsh margin at the top or the bottom of the scarp, as can be seen in Figure 3d.

Conversely, when grouped into change events, profiles exhibit a far greater degree of similarity, depicted by the latter three plots in each of the four groups in Figure 7. The distribution of $\Delta_{P,N}$ for change events of the same nature (retreat or progradation, respectively red and green dashed distributions) spans significantly lower values of $\Delta_{P,N}$ than that of $\Delta_{P,2}$ for paired profiles for change events of the same nature in all four instances shown in Figure 7. Furthermore, the distribution of $\Delta_{P,N}$ for pairs of change events of a different nature (grey dashed distributions) span values of $\Delta_{P,N}$ far greater than for profiles grouped by change events of the same nature (i.e., either progradation or retreat). The data therefore suggest that we can distinguish the morphology of marsh outlines affected or generated by change events of the same nature from those generated by different events, despite our inability to observe the morphology of the scarp itself. Akin to observations in mountainous

regions [60], we find that a key feature of salt marsh geomorphology, such as an erosion scarp, may be characterised at grid resolutions greater than its spatial dimension.

This observation alone does not imply an exclusive relationship between the nature of marsh outline mobility and the profile geometry observable through airborne lidar. As shown in Figure 6 (a,c), only a dozen change events of either retreat or progradation are larger than 1000 m$^2$ (0.1 ha) and for small events, the close proximity of profiles may play a role in their similarity. Hence, small events can skew the distribution of $\Delta_{P,N}$ toward lower values.

Figure 8 shows the relationship between $\Delta_{P,N}$ and various metrics relating to profile proximity. Panels (a,c) express $\Delta_{P,N}$ as a function of $D_{P,N}$ both before and after change events and show no clear relationship between the two quantities, with a 20-point moving median of $\Delta_{P,N}$ remaining relatively stable under 0.3 m for retreat and progradation events. $\Delta_{P,N}$ is also noted to be fairly constant with the surface area of change events (b,d). Both $D_{P,N}$ and $A_{CE}$ cause an increase in the number of profiles $N$: due to their regular spacing of 20 m, $L_P = 20 * N$ can be used to express the minimum length of the change event's seaward outline and also shows no clear effect on $\Delta_{P,N}$. From this we conclude that the distance between profiles exerts no clear positive or negative influence on $\Delta_{P,N}$, thus confirming that the similitude in geometry observed within change events is likely linked to the nature of their evolution.

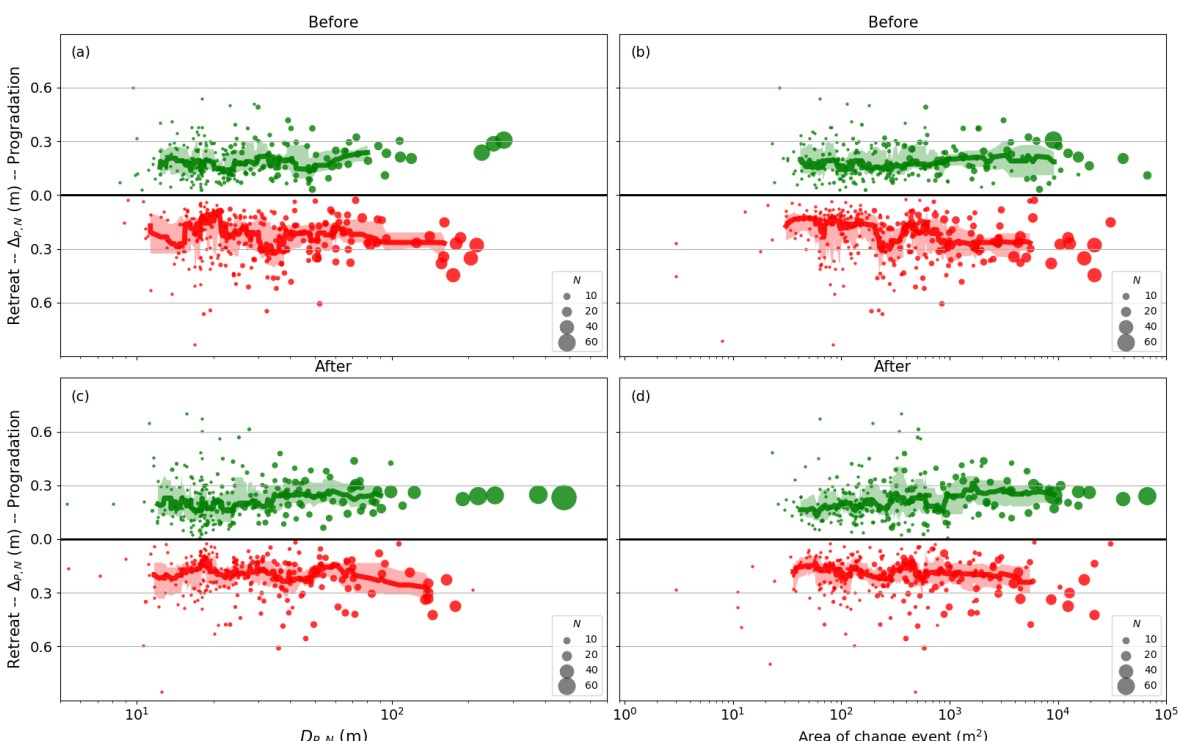

**Figure 8.** $\Delta_{P,N}$ for individual retreat and progradation events, expressed as a function of $D_{P,N}$ (**a,c**) and area of change event (**b,d**). (**a,b**) show profiles before events and (**c,d**) after events.

*4.3. Event Magnitude and Profile Morphology*

Having established that the different geometries of retreating and prograding marsh margins are observable from 1 m gridded lidar data, we investigate the influence of retreat and progradation on four topographic metrics: relief, scarp slope, marsh slope and mudflat slope. Figure 9 shows the distribution of the metrics within all sets of profiles that will undergo or underwent retreat or progradation between the identified periods of 2009–2013 and 2013–2017. Violin plots represent the distribution of $\Delta_{P,N}$ for the profiles described in the group. Coloured boxes in the boxplots show the interquartile range, with orange lines showing the median of the distribution. We show the median elevations of marshes and mudflats for each change event in Figure A5.

$R$ (see Equation (4)) ranges between 0 and 3.5 m and is noticeably larger for retreat events than progradation events at the same time step. This is in line with photographic evidence provided in Figure 1 and consistent with the hypothesis that progradation generates new low marsh platforms which accrete to elevations above the mudflat through time, thus getting more exposed to erosive factors and adopting the typical scarp morphology [26,61]. Both profiles about to be affected by change events and those generated by them appear to follow this pattern and also exhibit an increase in $R$ observed after change.

$Smax$ (see Equation (5)) follows a pattern similar to $R$ (this is inevitable given their definitions) but $Smax$ highlights the emergent patterns to a greater degree. On the other hand, contrary to $R$, $Smax$ is impacted by the resolution of the DTM. That retreating and prograding profiles show similar differences in $R$ and $Smax$ before and after change events suggests that a retreating profile is likely to conserve its shape and continue to retreat, as a prograding profile is likely to continue to prograde. However this statement appears contrary to the fact that $R$ and $Smax$ values associated to change in events in the 2013–2017 period begin lower than in the 2009–2013 period. We note that not all of the marsh outline is affected by change events in each period. Therefore, Figure 9 is not depicting a paradoxical decrease in relief between profiles generated by change events in 2013 and those affected by change events in 2017 but rather the two years' change events sample from a different distribution of profiles. This in turn suggests that the tendency of marsh outlines to continue evolving in their current direction may be reversed by external forcings more powerful than bank resistance, causing bank erosion.

The distributions of $S_{ma}$ and $S_{mu}$ (see Equation (6)) follow different patterns: $S_{ma}$ is consistently higher for progradating profiles than for retreating profiles. Indeed, retreating profiles often display a slope that dips toward the land rather than sloping offshore (e.g., the slope is negative in Figure 9c). This landward dip is likely due to higher deposition rates occurring close to creek networks and the marsh edge, predicted by models [62] and observed in the field [63]. This decrease in slope contrasts with the slight increase in $S_{ma}$ for prograding profiles after progradation. For progradation events, the age of the marsh platform before progradation is unknown. After progradation however, the marsh surface is only 4 years old. As shown previously [26,64], a young marsh platform is a transitional form closer to the original tidal flat than a fully developed marsh platform and therefore has a typically steeper slope. While we do not observe a significant difference in $S_{mu}$ between retreating and prograding profiles, we do note that retreating profiles experience an increase in mudflat slope after the retreat, whereas prograding profiles experience either no variation or a decrease in mudflat slope. These differences may be explained by the likelihood of a creek bordering retreated profiles which may cause the mudflat to steepen.

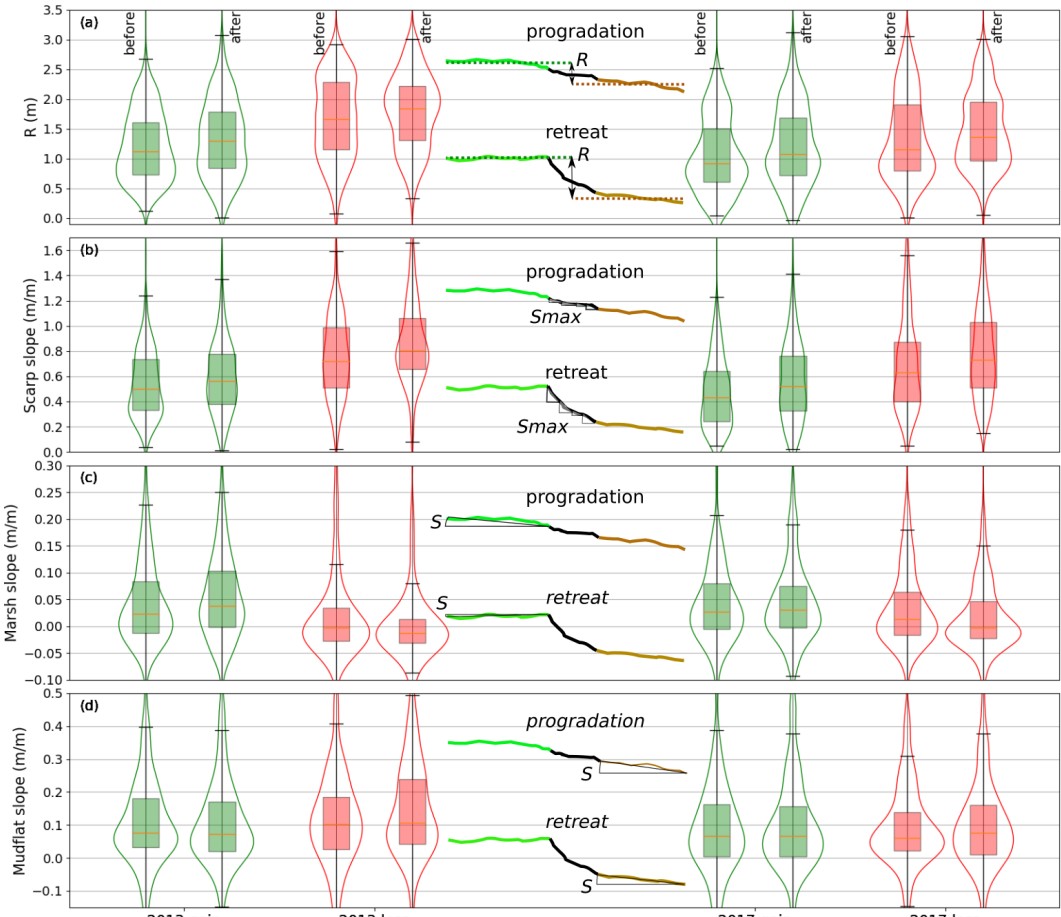

**Figure 9.** Boxplots and full distributions of marsh margin relief (**a**), maximum scarp slope (**b**), marsh slope (**c**) and mudflat slope (**d**). Diagrams in the centre of each panel represent the method to obtain the metric.

Figure 10 examines more closely the relationship between change event volume and *R*, which is the only metric depicted in Figure 9 that is independent of DTM resolution. We observe that relief tends to decrease with increasing progradation event volume, both before and after progradation. Therefore, large progradation events tend to affect marsh outlines with low relief and also generate new outlines with low relief. Notably, the largest progradation events are associated with a relief of less than 0.5 m. Hence, large progradation events produce marsh fronts which are close in elevation to the bordering mudflat. This creates a favourable environment for clonal and sexual colonisation, hydraulic conditions allowing [65]. This suggests that, barring variations of mudflat elevation, for example due to wind-waves [66,67], the marsh will prograde until hydraulic and chemical conditions are no longer suitable [65,68]. Conversely, relief shows no consistent trend with change event volume before retreat events, indicating that retreat may affect marsh outlines similarly regardless of their original relief (Figure 10a,c). However, after retreat events of more than 100 m$^3$ in 2013 and all retreat events in 2017, relief increases with change event volume. Retreat events of larger volume tend to increase relief because they remove platforms that are further inland and therefore generally higher.

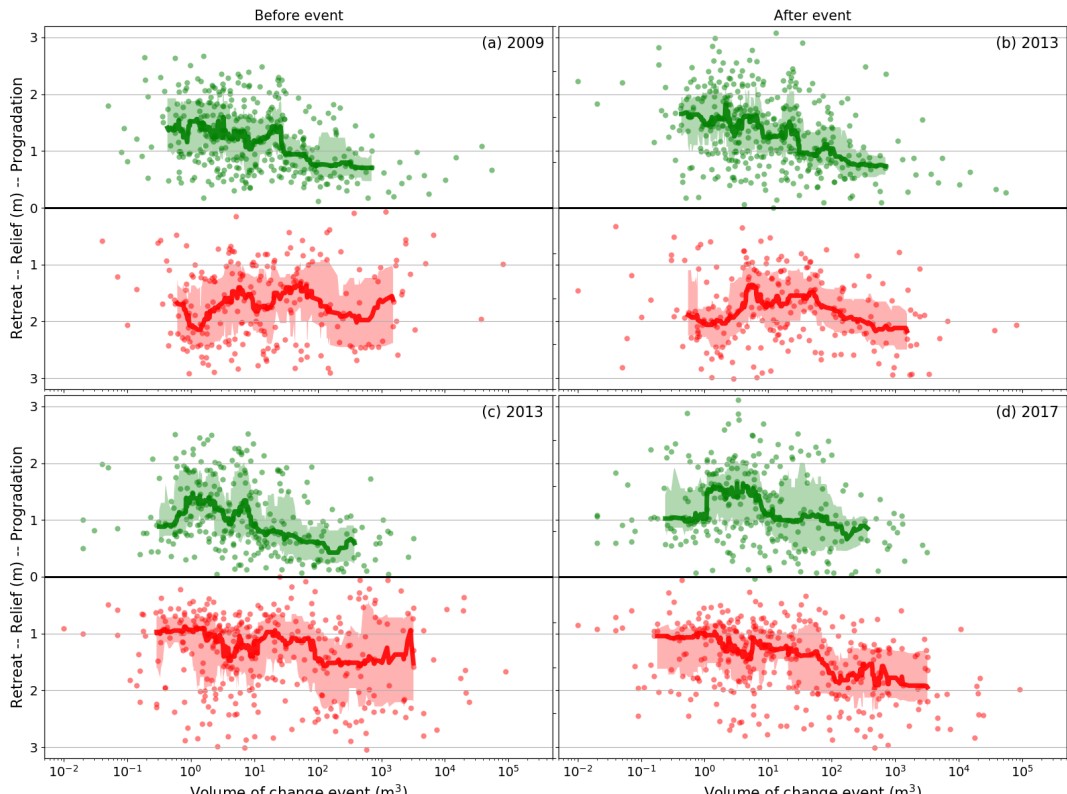

**Figure 10.** Marsh margin relief, expressed as a function of change event volume (the volume of loss events is negative) for profiles affected by change events (**a**,**c**) and resulting of change events (**b**,**d**), in the periods 2009–2013 (**a**,**b**) and 2013–2017 (**c**,**d**). Thick lines are a running median over 30 elements, surrounded by the 1st and 3rd quartiles (filled). Relief for prograding profiles (green) and retreating profiles (red) are mirrored through the $y = 0$ line.

### 4.4. Marsh Boundary Movement and Vertical Accretion

Figure 11 shows the relationship between the median initial marsh platform elevation $\tilde{p}_{ma,z}$ and the median change in $\tilde{p}_{ma,z}$ for profiles in individual change events between 2009 and 2013 (a) and 2013 and 2017 (b). We observe from the distribution of initial elevation that retreat events affect higher marsh platforms than progradation events and that change events between 2013 and 2017 affected lower platforms than in the 2009–2013 period. This result shows that during our study period, higher and therefore older or further upstream platform edges were more likely to undergo retreat. Concurrently, in both periods the decrease in $\tilde{p}_{ma,z}$ with initial elevation are very similar for retreat and progradation events. This implies that the rates of accretion at the platform edge are primarily controlled by their initial elevation rather than the direction of shoreline movement. The influence of initial elevation on accretion rates has been demonstrated before, notably using single-point models [46,69,70]. These models also emphasise the importance of suspended sediment concentration on accretion rates. Our results suggest that, for Moricambe Bay, sediment supply is not significantly larger near prograding platform edges than near retreating platform edges. Hence, we may reject the idea that heterogeneous sediment distribution in Moricambe Bay causes marsh platforms to prograde. Rather, the drivers of marsh edge evolution are external forcings such as tidal creek meandering that force retreat processes. Consequently, retreating platforms may prograde again as tidal creek thalwegs move away from them, as suggested by [36]. By extension, we infer that Moricambe Bay has sufficient sediment supply to support rapid infilling and conversion of the bay to marshes were it not for the action of meandering creeks.

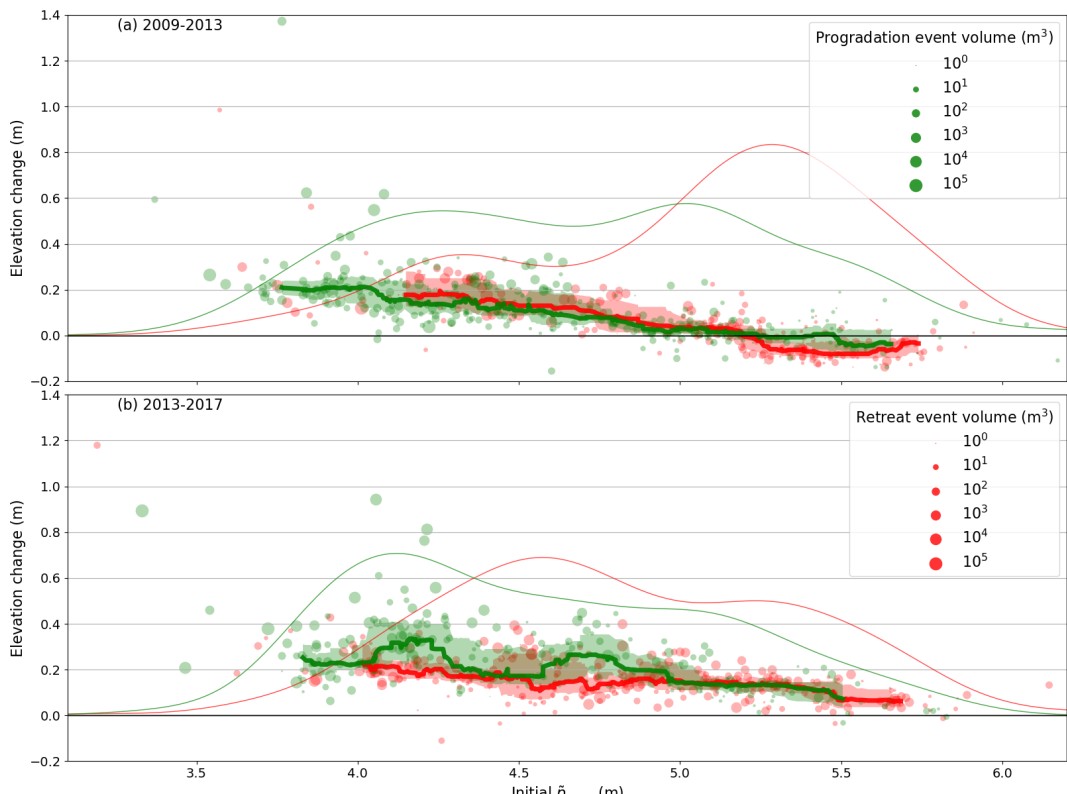

**Figure 11.** Vertical accretion of the marsh platform expressed as a function of initial platform elevation in the periods 2009–2013 (**a**) and 2013–2017 (**b**). Thick lines are a running median over 30 elements, surrounded by the 1st and 3rd quartiles (filled). Background red and green lines show the distribution of the initial elevation of change events.

## 5. Conclusions

In this contribution, we examine the morphological properties of both prograding and retreating salt marsh margins in Moricambe Bay, a sheltered mega-tidal bay for which topographic data are available at a grid step of 1 m and a vertical accuracy ranging from 3 to 7 cm. We use the TIP method [53] to determine the location of salt marsh margins for 3 surveys in 2009, 2013 and 2017. We then design and use a new algorithm to generate 10 m long topographic profiles, regularly spaced every 20 m along each margin. At the time of writing, we found very few studies focusing on the morphology and evolution of salt marsh scarps. While some seminal studies refer to marsh margins [71] and the bordering mudflats [72], they often define margins over several kilometres and ignore the meter scale structures that are scarps. This is, to our knowledge, the first analysis of salt marsh margins to cover a large marsh system at such high spatial resolution and the first to consider the variability of marsh margins in such close proximity to the marsh edge. We have used this dataset to determine whether marsh profile geometry before and after change events correlates with marsh profile evolution and to explore the evolution of simple metrics relating to profile geometry during retreat and progradation events.

We determine spatially contiguous change events (i.e., contiguous areas that have either prograded or retreated) and find that retreat events consistently have a lower surface-area-to-volume ratio than progradation events. That is, for a given area of marsh, a retreat event will excavate a larger volume of sediment compared to the volume of sediment deposited by a progradation event of the same surface area. This result, consistent with our field observations, suggests a morphological difference between retreating and prograding marsh margins. Hence, we analyse the spatial variation in profile geometry for both retreat and progradation events to see if profiles that prograde or retreat in the next timestep

are similar. Indeed, if prograding profiles were to look similar and not like retreating profiles, it could be possible to predict which parts of the marsh may retreat or prograde in the future.

We find that the difference between pairs of retreating or prograding profiles is not significantly lower than for randomly paired retreating and prograding profiles, precluding predictions for future evolution. However, we find profiles within change events to be similar to each other and different from profiles in other change events. We also find this similarity to be uncorrelated to the distance between all transects within a change event, implying that the observed pattern in profile geometry may be linked to marsh margin evolution processes.

A well-documented difference between retreating and prograding profiles is the presence of a sub-vertical scarp. Profiles that have retreated in the previous timestep have scarps, those that prograded do not. Having shown that there is an observable difference between retreating and prograding profiles despite the "invisibility" of scarps at the scale of observation, we proceed to explore four basic metrics of profile morphology. We find that the marsh-to-mudflat relief behaves similarly to the maximum observed scarp slope and is different for retreating and prograding profiles. In the absence of detailed observations of the scarp, we use this metric as a proxy for scarp height. We observe a noticeable difference between prograding profile marsh slopes, which dip seaward and retreating profile marsh slopes, for which landward dip increases after retreat. This suggests that retreating profiles are mainly observed in older terraces, whereas if left undisturbed, young prograding profiles will continue to prograde. Concurrently, we note that retreating and prograding scarps exhibit very close accretion rates of the marsh surface between time steps. From this we infer that accretion in our site is controlled by the initial elevation of the marsh surface to a greater extent than the loss or gain of marsh surface. This disconnection between vertical and horizontal growth shows that Moricambe Bay does not have a sediment supply deficit and confirms that in the absence of creek-driven erosion, marsh progradation would fill in the Bay.

This contribution highlights the richness of information that may be derived from a close examination of active marsh margins. This wealth has been partially uncovered by the availability of high-resolution lidar, however the limits of nadir-facing topographic data are strained for environments featuring complex sub-vertical structures such as erosion scarps. Previous work stresses the role of scarp geometry in determining wave thrust [32]. We suggest that future research in this field applies itself to oblique observations, as have been seen in morphological analyses of fault scarps [73], cliff faces [74] or river banks [75]. The resulting production of 3D models of marsh edges to better inform existing geomechanical models of scarp failure [34] and thus improve our predictions of marsh outline evolution.

**Author Contributions:** G.C.H.G. designed the study, wrote the software, performed the analysis and wrote the paper with inputs from S.M.M.

**Funding:** G.C.H.G. acknowledges funding from NERC grant NE/L002558/1. S.M.M. acknowledges funding from the Leverhulme Grant IAF-2014-009.

**Acknowledgments:** We acknowledge Marie Margaria for beta testing the software.

**Conflicts of Interest:** The authors declare no conflict of interest.

## Abbreviations

The following abbreviations and notations are used in this manuscript:

| | |
|---|---|
| CE | Change Event |
| DTM | Digital Elevation Model |
| DTM | Digital Terrain Model (A DTM of the ground surface) |
| DEFRA | UK Department for Environment and Rural Affairs |
| PE | Progradation Event |
| RE | Retreat Event |
| RMSE | Root Mean Square Error |

TIP       Topographic Identification of Platforms (a software package)
$A_{CE}$      Area of a change event
$V_{CE}$      Volume of a change event
$h_{CE}$      Average elevation change during a change event
$\tilde{X}$       the median value of a set X
$P$        A set of profiles
$pi$       The $i$th profile in a set
$pi_j$      The $j$th vertex of the $i$th profile in a set
$pi_{j,x}$     Distance to landward vertex of the $j$th vertex of the $i$th profile in a set
$pi_{j,z}$     Elevation of the $j$th vertex of the $i$th profile in a set
$p_{ma}$     the first 4 vertices in a profile
$p_{mu}$     the last 4 vertices in a profile
$\Delta_{P,N}$     Mean absolute difference in elevation between N profiles of a set P
$D_{P,N}$     Mean distance between N profiles of a set P
$R$        Relief: difference in elevation between
$Smax$    Maximum slope of the scarp
$S_{ma}$     Overall slope of $p_{ma}$
$S_{mu}$     Overall slope of $p_{mu}$

## Appendix A. DTM Offset and Ground-Truthing

*Appendix A.1. DTM Offset*

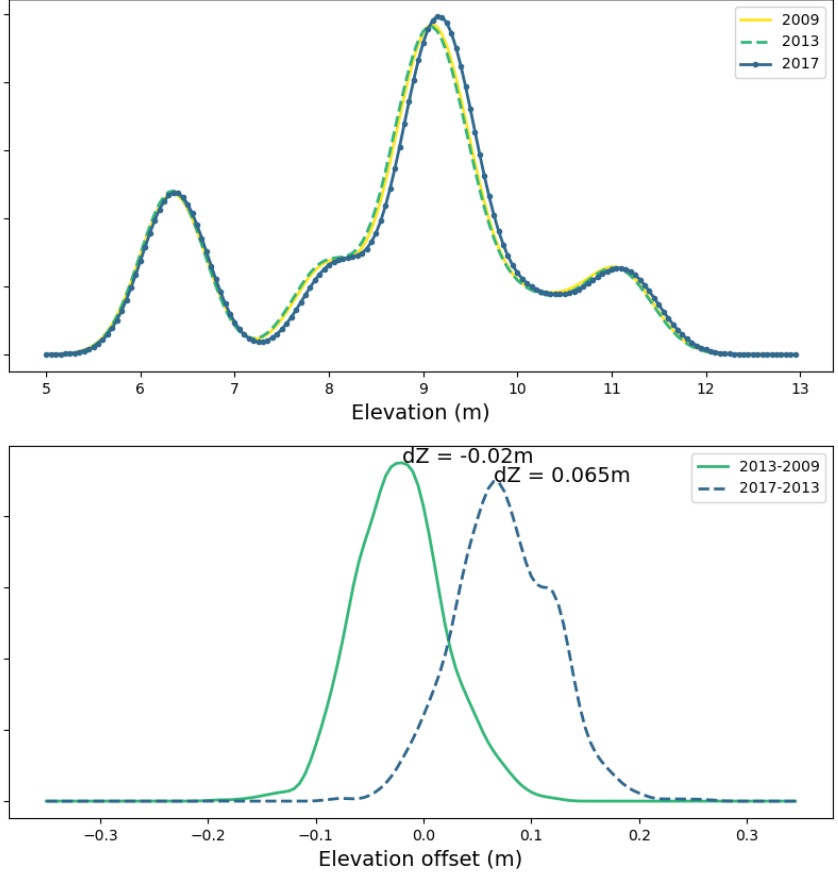

**Figure A1.** (**top**) Distribution of elevations for ground-truthing points in Moricambe Bay. (**bottom**) Distribution of elevation offset between DTM point elevations at the location of ground-truthing points at different dates.

*Appendix A.2. Ground-Truthing*

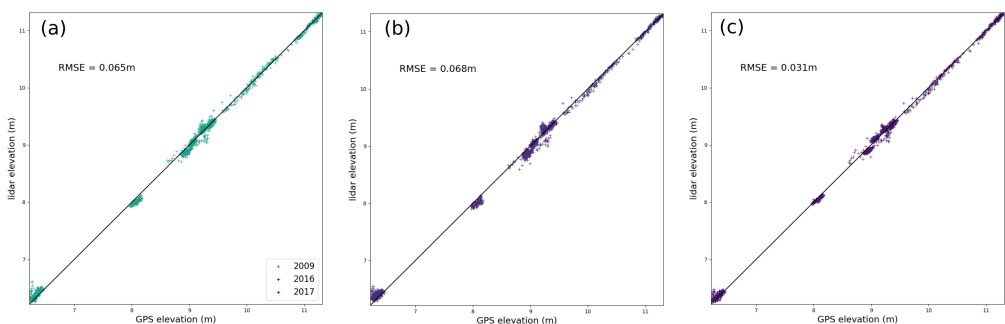

**Figure A2.** Comparative plot of elevations at ground-truthing points between the DTM and ground-truthing data of the same year or a close year. (**a**) the DTM year is 2009 and the ground-truthing year is 2009; (**b**) the DTM year is 2013 and the ground-truthing year is 2016; (**c**) the DTM year is 2017 and the ground-truthing year is 2017.

## Appendix B. Sectors and Parameters Used for the Tip Method

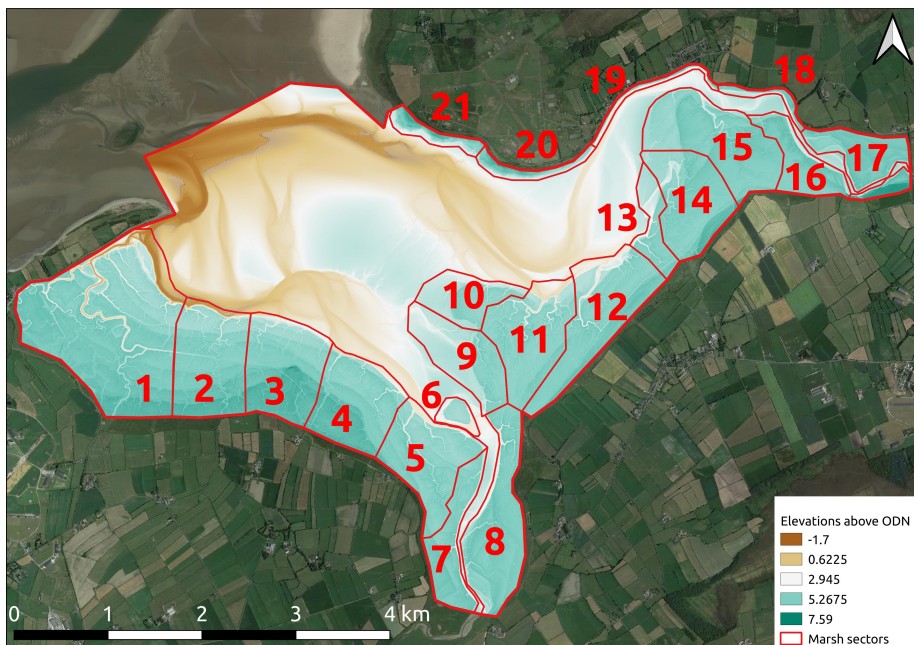

**Figure A3.** Map of the sectors used to implement the TIP method, overlain on the 2017 DTM of Moricambe Bay.

**Table A1.** The parameters used in the TIP method for each of the 21 sectors in the 2009 DTM.

| Sector | $Sp_{thresh}$ | $ZK_{thresh}$ | $rz_{thresh}$ |
|--------|---------------|---------------|---------------|
| 1 | −2.0 | 0.85 | 8 |
| 2 | −2.0 | 0.85 | 8 |
| 3 | −2.0 | 0.85 | 8 |
| 4 | −2.0 | 0.85 | 8 |
| 5 | −2.0 | 0.85 | 8 |
| 6 | −2.0 | 0.85 | 8 |
| 7 | −2.0 | 0.85 | 8 |
| 8 | −2.0 | 0.85 | 8 |
| 9 | −2.0 | 0.85 | 8 |
| 10 | −2.0 | 0.85 | 8 |
| 11 | −2.0 | 0.85 | 8 |
| 12 | −2.0 | 0.35 | 24 |
| 13 | −2.0 | 0.85 | 14 |
| 14 | −2.0 | 0.85 | 2 |
| 15 | −2.0 | 0.85 | 1 |
| 16 | −2.0 | 0.85 | 1 |
| 17 | −2.0 | 0.85 | 8 |
| 18 | −2.0 | 0.85 | 10 |
| 19 | −2.0 | 0.85 | 12 |
| 20 | −3.0 | 0.4 | 22 |
| 21 | −2.0 | 0.85 | 14 |

**Table A2.** The parameters used in the TIP method for each of the 21 sectors in the 2013 DTM. Stars (*) indicate manual modification of the marsh outline was performed.

| Sector | $Sp_{thresh}$ | $ZK_{thresh}$ | $rz_{thresh}$ |
|--------|---------------|---------------|---------------|
| 1 | −2.0 | 0.85 | 8 |
| 2 | −2.0 | 0.85 | 8 |
| 3 | −2.0 | 0.85 | 8 |
| 4 | −2.0 | 0.85 | 8 |
| 5 | −2.0 | 0.85 | 8 |
| 6 | −2.0 | 0.85 | 8 |
| 7 | −2.0 | 0.85 | 8 |
| 8 | −2.0 | 0.85 | 8 |
| 9 | −2.0 | 0.85 | 20 |
| 10 | −2.0 | 0.85 | 13 |
| 11 | −2.0 | 0.85 | 12 |
| 12 | −2.0 | 0.35 | 12 |
| 13 | −2.0 | 0.85 | 12 |
| 14 | −2.0 | 0.85 | 7 |
| 15 | −2.0 | 0.85 | 6 |
| 16 | −2.0 | 0.85 | 1 |
| 17 | −2.0 | 0.85 | 8 |
| 18 | −2.0 | 0.85 | 10 |
| 19 | −2.0 | 0.85 | 20 * |
| 20 | −3. | 0.4 | 22 |
| 21 | −2.0 | 0.5 | 12 |

**Table A3.** The parameters used in the TIP method for each of the 21 sectors in the 2017 DTM. Stars (*) indicate manual modification of the marsh outline was performed.

| Sector | $Sp_{thresh}$ | $ZK_{thresh}$ | $rz_{thresh}$ |
|--------|---------------|---------------|---------------|
| 1 | −2 | 0.85 | 8 |
| 2 | −2.0 | 0.85 | 8 |
| 3 | −2.0 | 0.85 | 8 |
| 4 | −2.0 | 0.85 | 8 |
| 5 | −2.0 | 0.85 | 8 |
| 6 | −2.0 | 0.85 | 8 |
| 7 | −2.0 | 0.85 | 8 |
| 8 | −2.0 | 0.85 | 8 |
| 9 | −2.0 | 0.85 | 16 |
| 10 | −2.0 | 0.85 | 10 |
| 11 | −3.0 | 0.5 | 13 |
| 12 | −3.0 | 0.5 | 13 |
| 13 | −2.0 | 0.1 | 10 |
| 14 | −2.0 | 0.9 | 4 |
| 15 | −2.0 | 0.85 | 6 |
| 16 | −2.0 | 0.85 | 3 |
| 17 | −2.0 | 0.85 | 1 |
| 18 | −2.0 | 0.85 | 10 |
| 19 | −2.0 | 0.85 | 30 * |
| 20 | −0.3 | 0.4 | 22 |
| 21 | −2.0 | 0.5 | 16 |

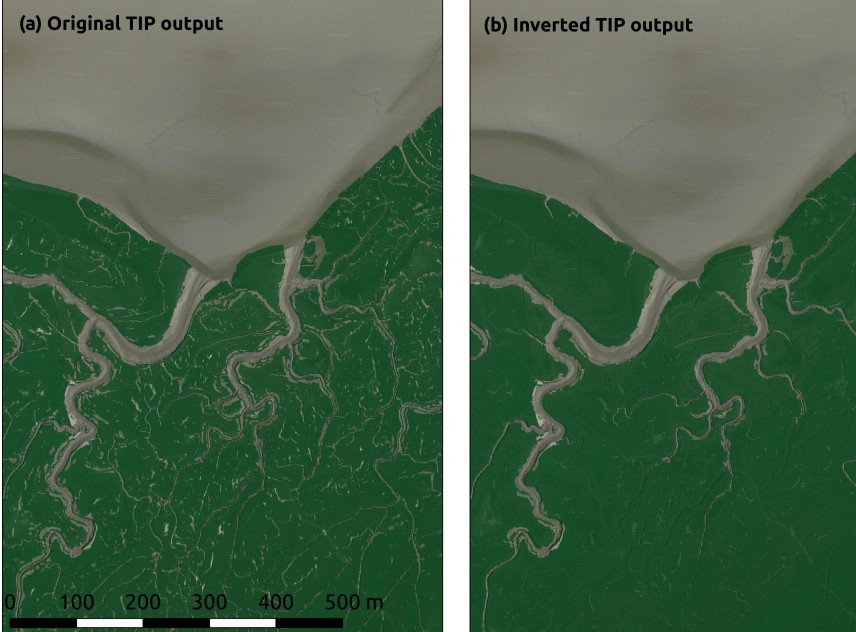

**Figure A4.** Example outputs of the TIP method, used in its original form (**a**) and "inverted" output, filled by considering as a marsh platform all pixels that are not part of the largest contiguous mudflat, in this case at the top of the panel (**b**). Marsh platforms are overlain over the Google Earth image of Figure 2.

## Appendix C. Raw Elevation Data

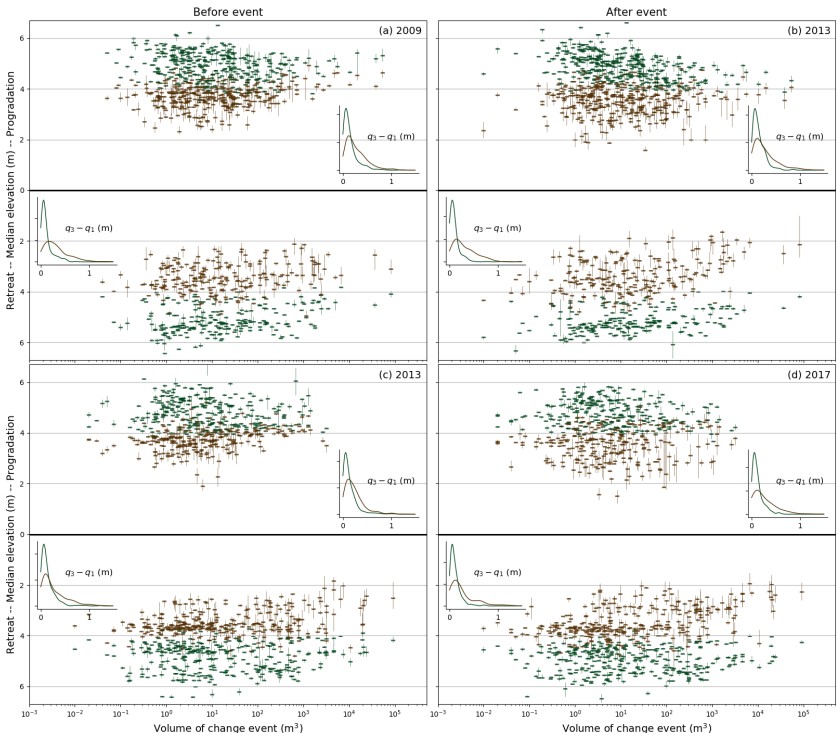

**Figure A5.** Median elevation (and surrounding quartiles) of the marsh (**green**) and mudflat (**brown**) portion of a group of profiles for individual change events. Progradation events are shown upward in each panel and retreat events are shown mirrored along the $y = 0$ line. Insets show the distribution of the interquartile range for marsh and mudflat portions of profiles.

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
