# Peer review of "Detecting the Morphology of Prograding and Retreating Marsh Margins—Example of a Mega-Tidal Bay"

_remotesensing, doi:10.3390/rs12010013_

Round 1

Reviewer 1 Report

In this study the authors used three years of 1m topographic data to investigate the evolution of Moricambe Bay, a sheltered mega-tidal bay. The TIP method was used to delineate salt marsh margins in each year of topography, and then profiles were generated along these margins to investigate morphological changes. Profiles were classified as experiencing retreat or progradation, and quantified in terms of their relief, slope (scarp, marsh, and mudflat), surface area, and volumetric change. These metrics were used to compare and contrast profiles before and after change events, and to ultimately determine the characteristics of Moricambe Bay’s evolution. Findings suggest retreat events tend to have lower surface-area-to-volume ratios, and that profiles experiencing similar events share morphological characteristics. Similarly, they found the presence of sub-vertical scarps indicative of retreating profiles, and suggest marsh-to-mudflat relief as a surrogate for identifying these features from topographic data, thereby overcoming an oft-cited limitation in the data.

The approach used is sound and the work is novel and important. However the overall structure of the paper needs some major re-organization prior to publication. There is considerable mixing of methods and results making the findings difficult to follow in parts. Further, the paper does not provide any information on the TIP method beyond a citation which is a central component of the entire analytical framework.

General Comments - see detailed line by line in the attached pdf

While the introduction provides a great background about evolving salt marshes, challenges to mapping their evolution, and the scope of this study, additional information would be welcome covering The role of salt marshes from a broader ecosystem function context and/or mitigation of sea level rise why salt marsh margins are important to accurately map, perhaps contextualized within future climate/sea level rise and/or management actions a bit more about the drivers of prograding and retreating of margin areas and how these may change with sea level rise/climate change Section 2.3 needs more details about the TIP method and how the sectors were selected. As written there is not enough context to evaluate whether the model parameterization and segmentation of the study area makes sense. The addition of a figure detailing the key morphological features of an idealized salt marsh margin profile would be helpful to familiarize readers with them and help contextualize the results. Perhaps including some idealized examples of retreat and progradation would be useful too. The choice of generating unique profiles for each DTM year makes interpretation of the results difficult. Tracking marsh evolution would be more easily interpreted if a uniform set of profile lines were used across all DTM years. The unique profiles could still be used to highlight morphology specific to the marsh front as determined from the TIP method for a given year, but a more uniform approach is warranted to facilitate the temporal discussion. This would also negate the complicated section on profile matching. The ‘Results and discussion’ section has a considerable amount of methodological material in it that should be moved to the methods section.

Author Response

We thank the reviewer for their insightful and encouraging comments. Please see below the modifications we made to our manuscript to take these comments into account. Bold text represents the reviewer's original comments and italic text is our response. Hopefully these modifications prove a satisfactory improvement of our contribution.

- additional information would be welcome covering the role of salt marshes from a broader ecosystem function context and/or mitigation of sea level rise why salt marsh margins are important to accurately map, perhaps contextualized within future climate/sea level rise and/or management actions a bit more about the drivers of prograding and retreating of margin areas and how these may change with sea level rise/climate change.

We have detailed the ecosystem services provided by salt marshes in the first paragraph of the introduction and described the potential consequences of their loss. We added the following sentences:
"Indeed, salt marshes participate in the filtration of organic and metallic pollutants and provide important nursing grounds for wildlife, including commercially exploited species such as Brown Shrimp.
Furthermore, their high productivity makes salt marshes important sequestrators of blue carbon, and vegetation and topography contribute to reducing storm surges and waves.
The loss of salt marshes to the sea is predicted to cause significant losses to the ecosystem services they provide and release stored carbon into the ocean, diminishing its capacity to siphon atmospheric carbon."

Likewise, we added the following to account for a better description of the importance of marsh scarp topography in retreat predictions:
"Multiple studies have focused on the impact of external forcing on the landward constriction of salt marsh habitat, as well as the mutual interaction between wave impact, retreat processes and the morphology of retreating marsh margins."

- Section 2.3 needs more details about the TIP method and how the sectors were selected. As written there is not enough context to evaluate whether the model parameterization and segmentation of the study area makes sense.
This is a highly pertinent observation. Initially we limited the details pertaining to the TIP methodology due to the availability of our previous publication on its design, parameters and limits. However, we agree that more detail on the choice of sectors is required for clarity, as well as on the working of the method itself. We detail our modifications to the manuscript in the line-by-line response.

- The addition of a figure detailing the key morphological features of an idealized salt marsh margin profile would be helpful to familiarize readers with them and help contextualize the results. Perhaps including some idealized examples of retreat and progradation would be useful too.

We modified Figure 1 to show diagrammatic marsh evolution under the typical morphologies seen in the photographs.

- The choice of generating unique profiles for each DTM year makes interpretation of the results difficult. Tracking marsh evolution would be more easily interpreted if a uniform set of profile lines were used across all DTM years. The unique profiles could still be used to highlight morphology specific to the marsh front as determined from the TIP method for a given year, but a more uniform approach is warranted to facilitate the temporal discussion. This would also negate the complicated section on profile matching.

We agree with the reviewer that our "unique profile" approach leads to a more complex interpretation than using one set of profiles for all three years. This is, however, a necessary precaution in our analysis: because marsh outlines do not prograde or retreat uniformly, a profile that is orthogonal to the 2009 marsh outline is unlikely to be orthogonal to the marsh outlines of 2013 and 2017. While in some cases the loss of orthogonality may not affect the observed scarp topography, more often than not it will result in an artificially increased tidal flat elevation and reduced tidal flat slope.
Therefore, while longer profiles would make a direct comparison between profiles easier, they also present the risk of generating misleading results.
Our modifications on the matter can be found in the line-by-line response.

- The ‘Results and discussion’ section has a considerable amount of methodological material in it that should be moved to the methods section.

We have moved this material to the 'Materials and Methods' section (see line-by-line response).

Response to individual comments (line numbers match first submission manuscript pdf):

l31: Lidar was spelled out in full.

l34: We modified the text to read: "in many micro-tidal systems and some meso-tidal systems the foot of the marsh scarp is rarely exposed, and few sites have as good topo-bathymetric data as the repeatedly studied Venice Lagoon in Italy, and Plum Island in Massachussets, USA, which are both the object of long-term monitoring campaigns"

l35: the text was modified to read: "Moreover, the spatial resolution of airborne lidar images is usually in the range of $1-5 m$, which reduces the perceived slope of scarps, despite being the most fine-grained remote sensing method used to cover large marsh systems \cite{Webb2013}. More importantly, scarps cannot be observed by nadir-facing airborne lidar surveys due to their sub-vertical face."

l47: the text was modified to: "Moricambe Bay (Cumbria, United Kingdom)"

l64: the text was modified to: "The bay of Moricambe, its North-West facing entrance enclosed between the Grune Cast sand spit and Cardurnock Flatts, is no exception and provides a sheltered environment where wide marshes have developed."

l71: the text was modified to: "The main activity on the salt marshes is cattle grazing, with both dairy cows and sheep regularly being kept in pastures on the marsh platforms. Hence, the dominant vegetation in Moricambe Bay is grazed".

Figure2: Labels were added

l78: the text was modified to: "We download point cloud topographic data from airborne lidar surveys of Moricambe Bay within the area of interest (red polygon in Figure \ref{fig:locmap} (a)) from the DEFRA data repository for 2009, 2013 and 2017 (\url{https://environment.data.gov.uk/DefraDataDownload/?Mode=survey}). DEFRA provides the last return for every point (the density of which does not exceed $6 \ pts \cdot m^{-2}$). This does not necessarily imply that the last return is the ground or bare earth, as dense vegetation on the marsh platform may prevent the laser from hitting the ground ."

l83: Here we refer to the mode, which signifies the point of the distribution with the highest frequency. We chose this metric as it is easily identifiable on the violin plots.

l92: the figures were modified accordingly

l97: We modified the first paragraphs to provide a summary of the process of the TIP method, including a description of the effect of the 3 parameters.

l99 and 101: this text was added: "The sectors were determined to minimise the overlap of mature and young platforms within any given sector, so as to avoid the TIP method mistaking the younger, lower platforms for tidal flats." A map of the sectors was also added in the appendix

l104: the text was modified to: "To keep only the most seaward outlines, we invert the TIP method's original results (see Figure A4 (a)) to identify tidal flats, of which we select only the largest. In Figure A4 (a), this is the northernmost tidal flat. Any pixel within the area of interest not classified as a tidal flat is then considered a marsh platform, yielding Figure A4 (b)."

l111: this text was added: "This is because the orientation of the marsh edge changes when the marsh outline progrades or retreats: hence, a profile that is orthogonal to the marsh outline in 2009 may not be in 2013 or 2017, thus rendering a direct comparison of profile geometry impossible.
An approach using sets of profile for each year is therefore preferable to one using a single set of profiles for all three years. Indeed, the latter approach, using longer profiles, would be suited to analyse the geometry of entire marsh platforms but not of features with small footprints like scarps. "

l136: the text was modified to: "Retreat events ($RE$), during which the marsh margin recedes landward, are lined with the most recent profiles on the landward side and the least recent on the seaward side, and vice versa for progradation events ($PE$).
Thus, each change event accepts as boundaries the marsh outlines that border it and is associated with two sets of profiles: one preceding the change and another resulting from the change (Figure 4(c))."

Figure 4: One of the other reviewers requested that panels (b,c) were made larger, and the other did not express an opinion on this figure. Hence, we elected to keep panel (c), which hopefully is made clearer by the amended text above.

Figure 5: Labels were added in panel (a) and the colour scheme entirely modified for better readability.

l161: the calculations were moved to the methods section

Figure 6: The scales were made to match in panels (a,c).

l184: the calculations were moved to the methods section

l207-209: Rather than being a by-product of the "unique profiles" approach, we believe this to be due to the diversity of profiles found in marshes and to the limited observation of their geometry using nadir-facing data. we added this detail to the explanation already present: "[...]DEM, and is in effect the difference in elevation between two contiguous pixels containing the scarp."

l234: the calculations were moved to the methods section

l249: the calculations were moved to the methods section

l292: While this would be a very interesting result (and has been examined by Cox et al (2003) and Butzeck et al (2016)), we feel that it is beyond the scope of this article. Future work may investigate the relationship between the distance to creeks and exerted shear stress on the banks, but would require a modelling component which we feel cannot feature in the present article.

l317: the paragraph was modified to: "[...] undergo retreat.Concurrently, in both periods the decrease in Pma,z with initial elevation are very similar for retreat and progradation events.
This implies that the rates of accretion at the platform edge are primarily controlled by their initial elevation rather than the direction of shoreline movement. The influence of initial elevation on accretion rates has been demonstrated before, notably using single-point models. These models also emphasise the importance of suspended sediment concentration on accretion rates. Our results suggest that, for Moricambe Bay, sediment supply is not significantly larger near prograding platform edges than near retreating platform edges. Hence, we may reject the idea that heterogeneous sediment distribution in Moricambe Bay causes marsh platforms to prograde. Rather, the drivers of marsh edge evolution are external forcings such as tidal creek meandering that force retreat processes.
Consequently, retreating platforms may prograde again as tidal creek thalwegs move away from them, as suggested byButzeck (2016)."

l319: We removed the notion of the Exner equation as we felt it no longer provide useful insight in the modified paragraph.

l372: We have now added mention of the lidar data in the introduction and methods.

Reviewer 2 Report

Dear authors,

this paper concerning retreat and progradation of edges of salt marsh platforms, demonstrated that marsh edges can be quantified with currently available DEMs (time-lapse) and oblique observations are crucial to fully describe scarps.

This paper is well written and real clear. 

I suggest you to improve in this way the figures:

fig.1 --> order of subfigures and text of these, and also improve the lines drawn 

fig.2 --> order of subfigures (invert a. and b.)

fig.3 and 4 --> zoom of subfigures b., c. and d. (only for fig.3)

fig.5 --> scale is lacking

fig. 7 --> increase text dimensions 

fig. 9 --> increase clearness of central diagrams

fig. 6, 8, 10, 11 --> increase the readability (data dimensions etc.)

Appendix

fig. A1 --> increase the readability changing type of lines (dotted, bold etc), please take in your mind that most of times the people print the paper in B&W

fig. A2 to A4 --> you can put all graphs in one figure

fig. A5 --> increase the text dimension and subgraph figure embedded 

Anyway, after these small corrections, I recommend this paper for publishing.

BR.

Author Response

We thank the reviewer for their pertinent comments and acknowledge that the original layout of our figures did not facilitate the comprehension of a relatively technical article.

We have modified most of our figures according to the reviewers' comments (in bold in the text below), and when this was not done we have provided an explanation for our choice (in italic in the text below). We hope these amended graphics provide the clarity expected. Further modifications were made to address the comments of other reviewers.

fig.1

order of subfigures and text of these, and also improve the lines drawn
The order of subfigures has been modified. We have not improved the resolution of the lines because it represents the resolution of our transects (this is added in the caption). We have, however, added a schematic of prograding and retreating profiles

fig.2

order of subfigures (invert a. and b.)
We did not change the order of subfigures to conserve a left-to-right and top-to-bottom order of the subfigures. The current configuration of the panels was chosen to maximise the dimension of panel (a), admittedly at the expense of panel (b), which we feel is of greater interest in this contribution. We have however added labels to panel(b) to provide context

fig.3 and 4

zoom of subfigures b., c. and d. (only for fig.3)
Fig.3 and 4 have been reordered to improve readability.

fig.5

scale is lacking
We added a scale bar and changed the color scheme according to the comments of reviewer 1 to improve readability.

fig. 7

increase text dimensions
This was done

fig. 9

increase clearness of central diagrams

The central diagrammes were increased in size as was the text. We hope these modifications added to the diagrammes inserted in Figure 1 sufficiently improve the clarity of our contribution

fig. 6, 8, 10, 11

increase the readability (data dimensions etc.)
We increased the text and label dimensions

Appendix fig. A1

increase the readability changing type of lines (dotted, bold etc), please take in your mind that most of times the people print the paper in B&W
We changed the colour scheme and added dotting on the lines

Appendix fig. A2 to A4

you can put all graphs in one figure
This was done

Appendix fig. A5

increase the text dimension and subgraph figure embedded

There was little space to increase subgraph dimensions, but we still increase this by a bit. We hope this is enough to make the figure readable.

Reviewer 3 Report

There are a lot parts of the ms that need significant improvement. Other parts remain cryptic and in several place there are jump in the paper logic. The used Δp,N or pmu , pma and other complicated symbols make the ms hard to the reader. The violin diagrams are not explained. The events referred in the text and in the figures are not explained and not appeared in the introduction. Although the ms is based on an excellent data base the end result is quite hard for the reader. 

Author Response

We thank the reviewer for their comments.

We understand the reviewer's concern that the manuscript is difficult to follow and have done our best to improve its flow. Namely, we have moved all of the formulas for our profile distance and profile topography metrics to the Methods section, which should address the logical jumps. We have also added explanations to read the violin plots in the text associated with Figures 7 and 9. To better announce the notion of change event detailed in Section 2.4 and Figure 4, we add a complement to the introduction. We are aware that our notation may be confusing, and have added these notations in the abbreviations and notations section. Furthermore, to improve the accessibility of the article, we modified Figure 1 to show diagrammatic marsh evolution under the typical morphologies seen in the photographs. Further modifications were made to address the comments of other reviewers.

Round 2

Reviewer 1 Report

Thank you for the responses to my comments. Your corrections addressed the majority of them and I feel the paper is stronger as a result.

Author Response

We thank the reviewer again for the helpful comments, and for accepting our contribution in its present state. We have performed an additional spell-check and addressed remaining grammatical issues.

Reviewer 3 Report

The new version of the manuscript is for sure imrpoved. I suggest however the authors to see some papers on the degradation modelling of scarps in active faults. These type of processes resembles your work because the formed step on the relief is progressively degraded soon after the event for example you can see the following publication KOKKALAS S., KOUKOUVELAS IK. (2005) Fault-scarp degradation modeling in central Greece: the Kaparelli and Eliki faults (Gulf of Corinth) as a case example. Journal of Geodynamics 40 (2-3), 200-215.   

Otherwise your paper is worth of publishing. Of course the paper is difficult but this caused because of the complexity of the geological problem.  

Author Response

We thank the reviewer for their final comments.

Indeed, our approach is close to monitoring applied in other domains, and we thank the reviewer for pointing out these synergies. To highlight the influence that other practices may have on marsh monitoring in the future, we modified the last sentence of the paper to read thus:

"We suggest that future research in this field applies itself to oblique observations, as have been seen in morphological analyses of fault scarps (Kokkala et al, 2005), cliff faces (Rosser et al, 2005) or river banks (Brodu and Lague, 2012).
The resulting production of 3D models of marsh edges to better inform existing geomechanical models of scarp failure (Bendoni et al, 2014) and thus improve our predictions of marsh outline evolution."

We hope this amendment satisfies your comment.